# Human Cadaveric Donor Cornea Derived Extra Cellular Matrix Microparticles for Minimally Invasive Healing/Regeneration of Corneal Wounds

**DOI:** 10.3390/biom11040532

**Published:** 2021-04-02

**Authors:** Arun Chandru, Parinita Agrawal, Sanjay Kumar Ojha, Kamalnath Selvakumar, Vaishnavi K. Shiva, Tanmay Gharat, Shivaram Selvam, Midhun Ben Thomas, Mukesh Damala, Deeksha Prasad, Sayan Basu, Tuhin Bhowmick, Virender Singh Sangwan, Vivek Singh

**Affiliations:** 1Pandorum Technologies Private Limited, Bangalore, Karnataka 560100, India; parinita.agrawal@pandorumtechnologies.in (P.A.); sanjay.ojha@pandorumtechnologies.in (S.K.O.); kamalnath.s@pandorumtechnologies.in (K.S.); vaishnavioptom@gmail.com (V.K.S.); tanmay.gharat@pandorumtechnologies.in (T.G.); selvam.shivaram@pandorumtechnologies.in (S.S.); ben.thomas@pandorumtechnologies.in (M.B.T.); 2Brien Holden Eye Research Center, LV Prasad Eye Institute, Hyderabad, Telangana 500034, India; mukeshdamala@lvpei.org (M.D.); deekshaprasad317@gmail.com (D.P.); sayanbasu@lvpei.org (S.B.); virender.sangwan@sceh.net (V.S.S.); 3Center for Ocular Regeneration (CORE), LV Prasad Eye Institute, Hyderabad, Telangana 500034, India

**Keywords:** decellularization, human cornea, extracellular matrix, fibrin hydrogels, in vivo imaging

## Abstract

Biological materials derived from extracellular matrix (ECM) proteins have garnered interest as their composition is very similar to that of native tissue. Herein, we report the use of human cornea derived decellularized ECM (dECM) microparticles dispersed in human fibrin sealant as an accessible therapeutic alternative for corneal anterior stromal reconstruction. dECM microparticles had good particle size distribution (≤10 µm) and retained the majority of corneal ECM components found in native tissue. Fibrin–dECM hydrogels exhibited compressive modulus of 70.83 ± 9.17 kPa matching that of native tissue, maximum burst pressure of 34.3 ± 3.7 kPa, and demonstrated a short crosslinking time of ~17 min. The fibrin–dECM hydrogels were found to be biodegradable, cytocompatible, non-mutagenic, non-sensitive, non-irritant, and supported the growth and maintained the phenotype of encapsulated human corneal stem cells (hCSCs) in vitro. In a rabbit model of anterior lamellar keratectomy, fibrin–dECM bio-adhesives promoted corneal re-epithelialization within 14 days, induced stromal tissue repair, and displayed integration with corneal tissues in vivo. Overall, our results suggest that the incorporation of cornea tissue-derived ECM microparticles in fibrin hydrogels is non-toxic, safe, and shows tremendous promise as a minimally invasive therapeutic approach for the treatment of superficial corneal epithelial wounds and anterior stromal injuries.

## 1. Introduction

Cornea is the transparent window of the eye and the most significant refractive element responsible for the creation of vision. However, at times, vision is compromised due to various factors including dystrophic, degenerative, infectious (bacterial and fungal infections), and secondary damage, such as scarring, chemical burns, and allergies, all of which if left untreated ultimately lead to irreversible loss of vision [1]. Among the various types of visual impairments, corneal blindness is the fourth leading cause of blindness globally [2]. On an average, 1.5–2 million new cases are estimated to occur worldwide each year [3]. In severe corneal injuries, the epithelium along with the stroma could be significantly compromised, threatening the structural integrity of the ocular surface. In such conditions, the most common treatment involves the replacement of a part or whole of the cornea with human donor tissue [1]. However, studies have shown that less than 5% of the patient population has access to corneal transplantation due to severe shortage of donor tissues and high cost of corneal transplantation [4]. Moreover, more than half of donated corneas do not meet the standards for transplantation due to low endothelial cell count, transmissible diseases, short shelf life, etc. Hence, other standard care options for treatment of corneal scars and perforation typically include application of tissue bioadhesives, use of artificial corneas, and biomaterial-mediated stem cell therapy [5,6,7,8].

Ocular tissue adhesives based on natural (fibrin, hyaluronic acid, collagen, and gelatin) or synthetic (cyanoacrylate- and PEG-based (ReSure and OcuSeal^®^) polymers have been extensively used as suture substitutes for closure of corneal incisions/perforation and conjunctival wounds [5]. For serious ocular complications, replacement of the diseased cornea with a synthetic “artificial” cornea has been actively investigated as an alternate option to a conventional corneal tissue graft [3,6,9]. In this regard, cell-free biomaterial implants composed of recombinant human collagen type III (RHCIII) or fabricated from poly(2-hydroxyethyl methacrylate) (AlphaCor^®^) have been evaluated on patients with severe corneal pathologies [3,6]. For instance, long-term assessment of patients with RHCIII implants demonstrated that the biosynthetic implants presented no serious adverse reactions, including pain or discomfort, excessive redness, and swelling of adjacent corneal tissues [3]. However, problems related to cytotoxicity from residual contaminants were one of the notable drawbacks of this approach [9]. Besides, cell-free systems are inherently limited by their capacity to homogenously recruit and integrate keratocytes from nearby stromal tissue which could severely impair biointegration and impede corneal stromal repair and regeneration in vivo. Hence, there is a demand for safer, efficacious, and cost-effective alternatives that would allow for successful treatment of corneal diseases in a minimally invasive manner.

Recently, the incorporation of extracellular matrix (ECM) components obtained from decellularized native tissues has gained importance as they offer tissue-specific biological cues that could stimulate host-cell migration, mediate stem cell differentiation, and offer pro-regenerative microenvironment conducive for tissue regeneration [10,11]. In this regard, decellularized freeze milled cornea powder has been reported for use in corneal repair and regeneration [12,13]. For instance, decellularized bovine and porcine corneas have been shown to be biocompatible and demonstrated the ability to maintain optical clarity, mechanical, and structural integrity favorable for transplantation procedures in vivo [14,15]. In addition, acellular porcine cornea matrix implanted in inter-lamellar stromal pockets in a rabbit model demonstrated excellent biocompatibility with optical transparency comparable to that of normal corneas eight weeks post-implantation [16]. In the present study, decellularized cornea ECM microparticles (dECM microparticles), derived from cadaveric human corneas that were unfit for transplantation, were dispersed in fibrin glue and their potential for facilitating cornea stromal wound healing was evaluated. To this end, dECM microparticles incorporated in fibrin hydrogels were characterized for their physical and mechanical properties in vitro. Additionally, fibrin–dECM hydrogel biocompatibility and its influence on human corneal stem cells (hCSCs) phenotype were demonstrated via fluorescence imaging in vitro. Lastly, dECM-dispersed fibrin hydrogels, with and without hCSCs, were evaluated in a rabbit model of anterior lamellar keratectomy [17] to demonstrate corneal re-epithelialization, stromal tissue repair, and integration with native tissue in vivo.

## 2. Materials and Methods

Study Design, Location, Duration, and Approvals: It was prospective study and conducted at LV Prasad Eye Institute, Pandorum Technologies Pvt. Ltd., Bangalore and Vimta Laboratories, in Hyderabad. The work was approved by the ethics committees (IRB and IC-SCRT, LV Prasad Eye Institute) of the respective institutes and animals were handled in accordance with the Association for Research in Vision and Ophthalmology (ARVO) statement for the animal use in Ophthalmic and Vision Research as well as in accordance with the tenets of declaration from Helsinki. The animal study was conducted after IRB approval from the ethics committee at LV Prasad Eye Institute and animal ethics committee of Vimta Labs Limited (Ethics Ref: LEC-12-19-372) (LVPEI-IC-SCR REF No 02-19-001).

### 2.1. Decellularization of Human Cadaveric Donor Corneas

The decellularization procedures were performed under mild chemical conditions with NaCl solution modifying previously published protocols [18,19] (Figure 1A). Briefly, human cadaveric corneas, unfit for clinical transplantation but suitable for research, were collected from Ramayamma International Eye Bank (LVPEI, Hyderabad, India) and processed in a cGMP facility. Corneas stripped of the epithelium and endothelium were cut using trephine (9.25 mm) and the resultant cornea buttons were washed with phosphate buffered saline (PBS) solution. They were later washed with betadine solution (1.5%, purified water) for 30 s, incubated in sodium chloride solution (NaCl, 1.5 M), and placed on a rocker at room temperature (RT) for 48 h with NaCl solution changed once every 24 h. The decellularized corneas were then incubated in PBS containing Pen-Strep (P/S, 100 U/mL Penicillin, 100 µg/mL streptomycin, and Lonza) for 24 h, following which they were treated in PBS containing DNase in an orbital shaker (200 RPM) for another 24 h at 37 °C. Finally, the corneas were incubated in PBS containing 2× P/S for 24 h at RT and were tested by immunostaining to make sure no cellular debris or foreign DNA was present in the tissue before further processing.

### 2.2. Processing of Decellularized Tissues

The decellularized corneas were washed three times with deionized water for 30 min at RT to remove salts and antibiotics. After washing, the wet corneas were cut into small pieces (<20 mm^3^) and transferred to the micro vial set of the freeze-miller. Of note, we have evidence to show that water present in the cornea pieces form icicles upon freezing which might act as a grinding agent during the milling process to yield fine microparticles (Appendix A). The cornea pieces were then allowed to pre-cool in liquid nitrogen for 5 min, following which they were freeze-milled in liquid nitrogen for 10 min (SPEX SamplePrep, USA, #6775-230). This cycle was repeated two more times to yield a finely milled cornea decellularized extracellular matrix (dECM) paste. The dECM paste was frozen at −80 °C and lyophilized in a freeze dryer (Labcogene, Scanvac CoolSafe Pro #110-4) 48 h. After lyophilization, the dECM was freeze milled again (3 cycles, 10 min each) to yield a finely milled powder that was weighed and stored at −80 °C until further use.

For enzymatic digestion, decellularized corneas were washed three times with deionized water, lyophilized for 48 h, and chopped into small pieces (<20 mm^3^). The cornea pieces were then freeze milled to yield dECM powder which was subjected to enzymatic digestion using pepsin immobilized on agarose beads (Protein A Agarose beads, #20333, Thermo-Scientific). Briefly, dECM (10 mg) was UV-sterilized (20 min) and dispersed in sterile acetic acid solution (0.22 micron-filtered, 8.5 M, pH 4.5). The concentration of the enzyme slurry used was ~0.025 mL slurry/mg of dECM. The digestion was carried out at 37 °C in 8.5 M acetic acid (Merck) at pH 4.5 for 72 h. Post-digestion, the beads were separated from the enzyme digested dECM (EdECM) suspension via centrifugation at ~200× *g* for 5 min. The final EdECM was dialyzed, and the resultant suspension was lyophilized to yield a fine, dry powder. Both physically processed and EdECM microparticles were used for various characterizations, as detailed in Table 1.

### 2.3. dECM Microparticle Characterization

#### 2.3.1. Scanning Electron Microscopy (SEM) with Energy Dispersive X-Ray Analysis (EDAX)

Physically processed and enzymatically digested corneal dECM microparticle (EdECM) size and morphology were recorded using a field-emission scanning electron microscopy (Zeiss Ultra-55, Oberkochen, Germany) operating at a voltage of 5 kV (Figure 2). Briefly, lyophilized physically processed or EdECM microparticles were sputter-coated with gold/palladium to achieve a 10-nm coating and visualized under low vacuum conditions. Elemental analysis by energy dispersive X-ray spectroscopy was also performed to determine molecular constituents of EdECM samples (Zeiss Ultra-55, Oberkochen, Germany).

#### 2.3.2. Thermogravimetric Analysis (TGA)

TGA analysis was performed on thermogravimetric analyzer (TGA Q500 V20.13, TA Instruments) at a heating rate of 10 °C/min from RT to 350 °C under nitrogen atmosphere.

#### 2.3.3. Dynamic Light Scattering (DLS)

For DLS measurements, dECM microparticles were dispersed in PBS (1 mg/mL), and the average hydrodynamic particle size and polydispersity index (PDI) were measured using a particle size analyzer (Litesizer 500, AntonPaar) at RT.

#### 2.3.4. Fourier Transform Infrared (FTIR)

The Fourier transform infrared (FTIR) spectrum of lyophilized dECM powder in the region of 400–4000 cm^−1^ was obtained using a FTIR spectrometer (Spectrum Two, Perkin Elmer) at RT.

#### 2.3.5. Sandwich ELISA Assay

Analysis of carryover pepsin from the immobilized pepsin digestion was verified using porcine pepsin quantitative sandwich ELISA kit (MyBioSource, San Diego, CA, USA) as per the manual provided by the manufacturer.

#### 2.3.6. Mass Spectrometry

Briefly, dECM microparticles were processed with 4% SDS lysis buffer and heated at 90 °C for 5 min. The treated sample was then sonicated for 3 cycles at 40% amplitude including intermittent resting for 1 min on ice between cycles. The samples were then sonicated at 12,000 rpm for 15 min at 4 °C and the supernatant was transferred to a fresh tube. After protein estimation, the samples (200 µg) were further processed with dithiothreitol (Merck), iodoacetamide (Merck), and chilled acetone (Merck). The resultant sample was centrifuged and the pellet was dissolved in urea (6 M). The dissolved sample was further subjected to trypsin digestion and LysC solution (Lys C protease in 50 mM triethyl ammonium bicarbonate buffer, Merck) treatment. The resultant peptide digest was acidified with 1% formic acid and cleaned using Sep-Pak C18 columns. The eluent was dried using a vacuum concentrator and analyzed in a mass spectrometer (LTQ Orbitrap Velos, Thermo Scientific).

### 2.4. Hydrogel Preparation

To prepare the working pre-gel fibrin solution, thrombin was reconstituted in calcium chloride solution (1 mL) to obtain a 500 IU/mL solution, and aprotinin solution (1 mL) was added to fibrinogen powder to yield 90 mg/mL solution (Tisseel Lyo, Deerfield, Baxter). The fibrinogen solution was kept under stirring at 37 °C to obtain clear liquid. For preparation of dECM based fibrin hydrogels, dECM microparticles were first dispersed in reconstituted thrombin solution and this solution was mixed with an equal volume of reconstituted fibrinogen solution. The mixture was allowed to gel at 37 °C for 20 min to obtain the liquid cornea hydrogel (Schematic representation in Figure 3A). The final dECM concentration in the resultant fibrin gel was 30 mg/mL (3% *w*/*v*). Hydrogels with dECM concentrations higher than 30 mg/mL were not considered as it was difficult to mix this dECM/thrombin mixture with highly viscous fibrinogen solution which severely affected the homogenous distribution of the microparticles in the fibrin glue. Fibrin hydrogels prepared in the absence of dECM microparticles was used as control gels.

### 2.5. Hydrogel Characterization

#### 2.5.1. Compressive Modulus

Cylindrical hydrogel discs were prepared using molds (6 mm diameter and 1 mm height) and were subjected to a crosshead speed of 1 mm/min and compressed to a maximum strain of 50% using BiSS mechanical tester (OmniTest 5kN with Vector Pro NT). The values for compressive strain (mm) and load (N) were recorded and the compressive moduli were calculated from the slope of the linear region between 0.1–0.2 mm/mm strain on the stress (kPa) versus strain (mm/mm) curves using BiSS software.

#### 2.5.2. Crosslinking Kinetics

The crosslinking kinetics of fibrin and fibrin–EdECM hydrogels was determined using a rheometer (Anton Paar Rheometer, MCR105). Briefly, pre-gel solution, comprising of 125 µL fibrinogen and 125 µL thrombin, was placed in between the parallel plate geometry (25 mm diameter) with 0.2 mm gap. The change in rheological behavior (storage modulus) of the hydrogels was then measured at constant a frequency of 1 Hz when subjected to an oscillatory load of 0.2% strain. 

#### 2.5.3. Ex-Vivo Burst Pressure

The sealing ability of fibrin-based hydrogels was evaluated on rejected human donor corneas. Briefly, a full thickness corneal wound was created by punching a hole (2 mm diameter) using a commercially available leather hole punch (Armor Heavy Rotary Leather hole punch, Visking). The cornea was then mounted on a Barron artificial anterior chamber (Katena, Denville, NJ, USA) and 15–20 µL of fibrin glue was applied to seal the hole. After gelation, air was gradually pumped into the system (2 mL/min) using a syringe pump (New Era Pump System Inc., Farmingdale, NY, USA, #NE-1600) and the pressure at which the hydrogel ruptured or delaminated from the cornea was recorded (PASCO wireless pressure sensor, #PS-3203) as the maximum burst pressure.

#### 2.5.4. MTT Assay on Hydrogel Extracts

MTT cell proliferation assay is a simple method for determination of viability of cells and was performed as per the International Standards (ISO) 10993-5 with six replicate samples. Briefly, fibrin and fibrin–dECM hydrogels were incubated in complete cell culture medium (0.2 g/mL Minimal Essential Medium (MEM) supplemented with 10% fetal bovine serum (FBS) and P/S) for a period of 24 h. Later, the hydrogel extracts and degradation products were diluted with MEM to yield test concentrations of 25%, 50%, and 75% and evaluated on L929 fibroblast cultures. For cell culture, L929 cells were seeded at 1 × 10^4^ cells/well in a 96-well plate in 100 μL of culture medium and incubated at 37 °C. After 24 h, media containing hydrogel extracts were added to complete cell culture medium at the concentrations mentioned above. After 24 h incubation, MTT solution (HiMedia, India; 50 µL, 1 mg/mL) was added to each well and the plate was incubated for 2 h at 37 °C. The resulting formazan was then solubilized and quantified spectrophotometrically at 570 nm using an automated microplate reader (Perkin Elmer, EnSpire^®^, Waltham, MA, USA).

#### 2.5.5. Bacterial Reverse Mutation Test on Hydrogel Extracts

To study the genotoxicity of hydrogel extracts in vitro, a bacterial reverse mutation assay was performed using *S. typhimurium* and *E. coli* tester strains as per ISO 10993-3. For preparation of hydrogel extracts, fibrin and fibrin–dECM hydrogels (0.2 g/mL) were incubated under agitation in polar (0.9% *w*/*v* NaCl) and non-polar (sesame oil, MP Biomedicals) solvents at 50 °C for 70 h. Then, the mutagenicity of polar and non-polar hydrogel extracts was assessed based on the results of precipitation test and initial cytotoxicity test. For the mutation assay, plate incorporation and pre-incubation methods were carried out with 100% extract solution/plate, as the maximum limit of concentration under study groups along with solvent controls and positive controls, as mentioned in Appendix A. In the plate incorporation method, the tester strains along with induced rat liver S9 homogenate in PBS (used as the metabolic activation system) were mixed directly with 2 mL soft agar containing histidine-biotin for *S. typhimurium* and tryptophan for *E. coli* and poured on to minimal glucose agar plates (HiMedia). In the pre-incubation method, the strains along with S9 homogenate in PBS were incubated in a shaker at 100 RPM at 37 °C for 30 min prior to mixing with agar. Later, 100 µL of hydrogel extract were plated with each of the following tester strains: *S. typhimurium* TA98, TA100, TA1535, TA1537 and *E. coli* WP2 uvrA (pKM101) (Molecular Toxicology Inc., Boone, NC, USA), with and without metabolic activation, and the plates were incubated at 37 °C for 65 h. The condition of the bacterial background lawn was evaluated for evidence of hydrogel extract cytotoxicity using a code system and revertant colonies for a given strain exposed to different dilutions were counted manually.

### 2.6. In Vitro Cell Culture Studies with hCSCS

#### 2.6.1. Human Corneal Stem Cell Harvest and Expansion

Human corneal stem cells (hCSCs) were isolated and expanded as per our previously published protocol [20]. Briefly, hCSCs were isolated from collagenase digested limbal stromal tissue of human corneal rims from which central tissue had been removed. The isolated cells were cultured in Advanced DMEM/F12 medium containing 2% fetal bovine serum, epidermal growth factor (10 ng/mL), platelet-derived growth factor (PDGF-BB, 10 ng/mL), and gentamicin (50 ng/mL). Once confluent, hCSCs were passaged by trypsinizing with TrypLE and re-seeded at a density of ~10^4^ cells/cm^2^. The cells were cultured for up to three passages and used at passage four for subsequent experiments. On average, one cornea rim provided ~8–10 million hCSCs at passage 3 and around 3–5 corneas were used for in vitro cell culture studies.

#### 2.6.2. Cell Encapsulation in Hydrogels

Briefly, EdECM microparticles were UV sterilized inside a biosafety cabinet for 30 min and were suspended in reconstituted thrombin solution containing hCSCs. Next, the cell/EdECM suspension in thrombin was pipetted onto the culture surface and mixed with an equal volume of reconstituted fibrinogen solution. The mixture was allowed to gel at 37 °C for 20 min after which culture medium was added to the gel. The cell encapsulated hydrogels were then incubated at 37 °C in a 5% CO_2_ incubator. The final EdECM concentration in fibrin glue gel was 30 mg/mL (3% *w*/*v*) with a final cell density of ~1.5–2.0 million cells/mL of the hydrogel. 

#### 2.6.3. Live/Dead Assay

Briefly, thin hydrogels encapsulating hCSCs (2 × 10^6^ cells/mL) were prepared by placing a drop of hydrogel formulation (12 µL) at the center of a sterile coverslip. Then, another coverslip was placed immediately on top of the drop to yield a hydrogel sandwiched between the coverslips. The sandwiched hydrogel was then placed in culture media for a few minutes and then the top coverslip was gently released and pushed aside using a pair of sharp pointed forceps without damaging the underlying hydrogel. The hydrogels were cultured for a period of 5 days, following which they were incubated in calcein acetoxymethyl (calcein-AM, 0.2 µg/mL) and ethidium homodimer (2.5 µg/mL) (Invitrogen, Paisley, UK) for 15 min in supplemented DMEM at 37 °C to stain for live cells and dead cells. Live cells were visualized as green and dead cells as red under a fluorescence microscope (EVOS FL Auto2, Thermo Fisher Scientific, Waltham, MA, USA).

#### 2.6.4. Biomarker Expression

Briefly, fibrin derived hydrogels with encapsulated hCSCs (12 µL) were prepared between cover slips to yield thin samples. The thin hydrogel samples were recovered at different time points and washed with PBS. After washing, the hydrogel samples were fixed with 3.7% formalin solution in PBS (Sigma-Aldrich, India) for 10 min at RT and washed with PBS again. Immunohistochemistry (fluorescence) was performed to visualize the expression of scarless-wound healing related markers. Primary anti-CD73, anti-CD90 and anti-alpha smooth muscle actin (α-SMA) antibody were procured from Abcam. Goat anti-mouse and anti-rat antibodies (conjugated with Alexa Fluor 594 and Alexa Fluor 488; Thermo Fisher Scientific) were used as secondary antibodies. Briefly, after washing, the hydrogel samples were blocked in BSA (5% in PBS, 1 h) and were incubated with primary antibodies at 4 °C overnight (1:100 dilution in 1% BSA). The samples were then washed (PBS, 3 times, 5 min each) and incubated with an appropriate secondary antibody (1:200 dilution in 1% BSA) at room temperature for 1 h. The sections were then washed (PBS, 3 times, 10 min each), semi dried, and mounted with Vectashield antifade mounting medium containing DAPI (#H1200, Vector Labs, Burlingame, CA, USA). The sections were imaged under a fluorescence imaging system (Evos FL Auto 2).

#### 2.6.5. Biodegradation Study In Vitro

To assess in vitro biodegradation, hydrogels loaded with hCSCs (12 µL, 2 × 10^6^ cells/mL) were cultured for a maximum period of 9 days. At various time points, hydrogel samples were collected, lyophilized, weighed, and the mass remaining was calculated using the formula below:(1)$$\%\;mass\;remaining=\;\frac{\left(Mo-Mt\right)\;*\;100}{Mo}$$
where *Mo* is the initial dry mass of the hydrogel and *Mt* is the dry mass of the hydrogel at various time points.

### 2.7. In Vivo Studies

Skin sensitization test in Guinea pigs and acute ocular irritation test in rabbits were performed in an AAALAC accredited facility in accordance with the recommendation of the Committee for the Purpose of Control and Supervision of Experiments on Animals (CPCSEA) guidelines for laboratory animal facility published in the Gazette of India, 2018, in accordance with the protocol approved by Institutional Animal Ethics Committee (IAEC) (Protocol Nos. BIO-IAEC-3579 and BIO-IAEC-3385) and in accordance with the International Standard ISO 10993 Second Edition: 2006-07-15, “Biological Evaluation of Medical Devices—Part 2: Animal Welfare Requirements” (Reference Number: ISO 10993-2:2006(E)).

The corneal stromal injury in rabbit model was conducted after IRB approval from the ethics committee at LV Prasad Eye Institute and animal ethics committee of Vimta Labs Limited in accordance with the tenets of declaration from Helsinki (Ethics Ref: LEC-12-19-371).

#### 2.7.1. Skin Sensitization Test in Guinea Pigs 

For the study, animals were divided into the following four groups with at least 5 animals per group: polar solvent control extract, polar hydrogel extract, non-polar solvent control extract, and non-polar hydrogel extract, prepared using the method described in Section 2.5.5. The polar and non-polar extracts were then injected intradermally during induction phase (Day 1) and applied topically during boosting (Day 8) and challenge (Day 22) phases. For intradermal injections, animals received 0.1 mL of injection at the shoulder region, and, for topical application, filter papers soaked with polar or non-polar extracts were placed as a patch at the test site for 24 h. Animals were then observed for at least 24 h to see whether the extracts produced any skin reactions. In addition, all animals were observed once daily for clinical signs of toxicity and twice daily for mortality. Body weight was recorded prior to initiation of the treatment (Day 1) and at termination of study.

#### 2.7.2. Acute Ocular Irritation Test in Rabbits

The ocular irritation potential of fibrin and fibrin–dECM hydrogel extracts was evaluated via acute ocular irritation test in three New Zealand White rabbits. Hydrogel extracts were prepared using the method described in Section 2.5.5. The study was performed in two phases, initial test with one animal and confirmatory test with two animals. For both tests, a volume of 0.1 mL of undiluted extracts was instilled into the lower conjunctival sac of the eye and eyes were examined for ocular reactions at 1, 24, 48 and 72 h after instillation. At each interval, the cornea, iris, and conjunctivae were examined and scored according to a numerical scoring system. All three animals were observed once daily for clinical signs of toxicity and twice daily for mortality throughout the study period. Body weight was recorded on the day of instillation (Day 1 prior to instillation) and at termination (Day 4) of the experiment.

#### 2.7.3. Treatment of Corneal Stromal Injury in Rabbit Model

Nine 8–10-week-old New Zealand White rabbits weighing 2–3 kg each were chosen for the corneal wound healing study and acclimatized for at least 1 week. Wound model was created as per our previously published literature [17]. Briefly, rabbits were anesthetized using ketamine and xylazine, after which they were shifted to a surgical table and draped for surgery. The wound creation was performed on the left eye at the central zone of the cornea with the help of a skin-marking pen and trephine (Joja Surgicals Private Limited, Kolkata, West Bengal, India). Then, a guarded knife (400 µm) and crescent blade (Joja Surgicals) were used to make the corneal wound (3 mm dia, 150–200 µm depth), after which an Alger brush was used to clear the remnant corneal pieces. Animals were divided into the following four groups. These were untreated control (2 rabbits), fibrin (2 rabbits), fibrin + EdECM (30 mg/mL) (3 rabbits), and fibrin + EdECM + hCSCs (5 × 10^6^ cells/mL) (2 rabbits). Animals received one dose of ~4–8 µL of the formulation based on wound dimensions and clinician’s discretion. Lastly, a soft bandage contact lens (Purecon Lenses Private Limited, New Delhi, India) was placed and tarsorrhaphy was performed to ensure that the hydrogels were properly secured in the eye. Animals that underwent surgery but received no treatment were used as untreated controls and the healthy contralateral eyes of all rabbits were used as normal experimental controls. The schematic diagram of the procedure is shown in Figure 4A. All animals were treated with corticosteroids and antibiotics for a week post wound creation.

Rabbit corneas were imaged using standard protocols pre-surgery and at 1, 2, 4 and 8 weeks postoperatively (±1 day). To monitor epithelialization, a digital SLR camera (Nikon D3300) and a handheld slit lamp (PSLAIA-11, Appasamy Associate, Chennai) with blue filter were used to photograph and capture fluorescein-stained images of the cornea. Anterior segment optical coherence tomography (ASOCT) imaging was performed using Ivue (Optovue, USA) to visualize cross sections of the cornea to discern scar depth, corneal thickness, and corneal edema. Scheimpflug imaging was performed using Galilei G4 (Zeimer, Switzerland) in both horizontal and vertical meridians to analyze corneal topography and densitometry. 

### 2.8. Histopathology

After euthanasia at 8 weeks, rabbit corneas were punched out using 14 mm trephine and preserved in formalin. They were later cut into two halves, dehydrated, embedded in paraffin wax, and sectioned at a thickness of 4 µm. The cornea sections were then deparaffinized at 70 °C for 2 min and dipped in xylene solution immediately. After a series (100%, 90%, and 80%) of xylene wash, the sections were air dried and were subjected to an alcohol gradient wash (100%, 90%, and 80%) for 5 min each. Later, the slides were rehydrated in water for 10 min and stored in PBS. The deparaffinized sections were then stained with hematoxylin and eosin (H&E) and periodic acid-Schiff (PAS) staining respectively, as per standard protocols [21]. After completion of the staining protocols, the sections were mounted with DPX mounting media for imaging.

### 2.9. Immunofluorescence Staining

Heat induced antigen retrieval was performed on deparaffinized sections placed in sodium citrate buffer (pH 6.0). The sections were then washed (PBS, 3 times, 5 min each) and incubated in a permeabilizing agent (0.3% triton-X100, 30 min). After washing, the sections were blocked in BSA (2.5% in PBS, 1 h) and incubated with primary antibodies, cytokeratin 3 (CK3, 1:200) (SC-80000, Santa Cruz Biotechnology, USA) at 4 °C overnight, and alpha smooth muscle actin (α-SMA, 1:600) (A2547, Sigma-Aldrich, USA) at room temperature for 2 h. The sections were then washed (PBS, 3 times, 5 min each) and incubated with an appropriate secondary antibody (1:500 dilution in 1% BSA) (A11005, Invitrogen, USA) at room temperature for 1 h. The sections were then washed (PBS, 3 times, 10 min each), semi dried, and mounted with Fluoroshield medium containing DAPI (ab104139, Abcam, UK). The stained sections were then imaged under a fluorescent microscope (Axio Scope A1, Carl Zeiss AG, Germany) with 20–40× objective.

### 2.10. Statistical Analysis

Data are represented as mean ± standard error (SE) with n ≥ 3 samples/group. A two-tailed, unpaired Student’s *t*-test or Multiple *t*-test was performed as applicable to detect differences between two groups with *p* values ≤ 0.05 considered statistically significant.

## 3. Results and Discussion

Globally, millions of people suffer from corneal blindness and the most common treatment option for these patients is a corneal transplantation. However, due to acute shortages of donor corneas, only one in 70 patients benefits from it [4]. Furthermore, as approximately one-third of corneal transplantations end up in graft rejection or revision surgery in the long term [22], there is a clinical need for exploring other therapeutic avenues for patients with corneal pathologies. Although options such as the use of artificial corneas for restoring corneal clarity are employed in some cases [3,6,9], these prefabricated implants fundamentally require the need of a certified surgeon with surgical skills and advanced instrumentation for corneal surgery. In this study, we developed a minimally invasive methodology combining a biocompatible tissue adhesive, fibrin glue, and decellularized corneal ECM microparticles for treatment of superficial corneal epithelial defects and anterior stromal disorders. We hypothesized that the decellularized biological components, which possess structural and biochemical similarities to that of native microenvironment, should aid in homeostasis and regeneration of neo-tissues in vivo [10,11,23]. To this end, discarded human cadaveric donor corneas that screened negative for infectious diseases and viruses were decellularized, milled, and enzymatically digested to yield microparticles for ocular surface reconstruction.

### 3.1. Decellularization of Cadaveric Human Corneas

Decellularization of cadaveric donor corneas was performed under mild chemical conditions with hypertonic NaCl solution as this methodology yielded corneal tissues with minimal loss in transparency and stromal disruption [18,19] (Figure 1A). Of note, NaCl-based decellularization is biocompatible, compared to techniques that involve ionic/non-ionic detergents, as it relies only on osmotic shock to trigger cell rupture and cell death without significantly altering the structural integrity of native cornea tissue [18]. The resultant acellular corneal scaffolds obtained after NaCl treatment were characterized and evaluated to ensure the completeness of the decellularization protocol. The decellularization efficiency of the human corneal tissues was confirmed via H&E and DAPI staining (Figure 1B). Cross-sections of stained decellularized tissues demonstrated the absence of remnant cells and DNA. These results show that the employed NaCl-based decellularization protocol efficiently removed all cell debris and associated genetic material while preserving the structural, biochemical, and biomechanical cues of the corneal ECM structure.

### 3.2. Characterization of Cornea Derived dECM Microparticles

After decellularization, the cornea buttons were incised and milled in a freeze miller to yield particles (Figure 1C) that ranged 7–10 µm. More notably, the physically processed dECM microparticles were found to be non-homogenous and had the tendency to form large agglomerates over time (Figure 2A). As high moisture content in samples promote agglomeration and is often correlated to poor stability under long-term storage conditions [24], we determined the moisture content in the freeze-milled samples using thermogravimetric analysis (Appendix A). TGA data demonstrate that the freeze-milled dECM microparticles possessed ~7.7% moisture content at 100 °C which increased to 9.5% at 200 °C (Appendix A). To circumvent the issue of microparticle agglomeration and yield particles of smaller size with homogenous particle distribution for easy dispersion in fibrin glue, we explored the possibility of enzymatically digesting the dECM powder with porcine pepsin immobilized on agarose beads (Figure 1C).

SEM data show that the resultant enzymatically-digested corneal dECM microparticles averaged <4 µm in size, which was a two-fold decrease in particle size compared to physically-milled dECM powder (Figure 2B). Furthermore, EdECM particles were mostly found to be spherical in shape with characteristic petaloid-like architectures (Figure 2C). This observation was unique and is considered advantageous as these ultrastructures could increase the availability of cell binding RGD epitopes and facilitate cell–ECM interactions at the macroscopic level [25]. In addition, elemental analysis of EdECM microparticles confirmed its ECM origin (not inorganic or salt particles) as it was primarily composed of carbon and oxygen moieties (Figure 2D). The size of EdECM microparticles dispersed in aqueous solution was evaluated using dynamic light scattering. DLS measurements demonstrated a homogenous particle distribution with polydispersity indices <0.3 and particle sizes <1 µm (Figure 2E).

To demonstrate that enzymatic digestion did not modify or alter the ECM protein structure, lyophilized dECM microparticles were analyzed using FTIR spectroscopy. The results demonstrate that FTIR spectra of both physically processed and EdECM microparticles exhibited characteristic amide peaks [26], commonly seen in biological proteins, with no significant differences observed between them (Appendix A). The C=O stretching vibration in the amide group of the dECM protein was seen at 1630 cm^−1^, while the out-of-phase and in-phase combination of N-H bending and C-N stretching vibration were observed at 1542 and 1403 cm^−1^, respectively [26].

Xenogenic components have the tendency to elicit strong immune responses compared to allogenic materials in vivo [27]. To rule out the possibility of porcine pepsin in EdECM microparticles, a sandwich ELISA assay was performed on lyophilized EdECM microparticles (Appendix A). Our ELISA results demonstrate an insignificant amount of porcine pepsin (0.0001%) present in the dECM powder digested with immobilized pepsin under mild acidic conditions. Next, to demonstrate that most corneal dECM components were retained after immobilized pepsin treatment, mass spectrometric analysis was performed on the EdECM powder (Appendix A). MS results show that majority of corneal ECM components, including collagen I, collagen V, and cornea-specific proteoglycans, such as keratocan, decorin, lumican, and biglycan, as reported previously [28], were preserved in the pepsin-digested dECM powder. More importantly, MS analysis established the absence of run-away porcine pepsin in the final EdECM digest. Cumulatively, these observations indicate that enzymatic digestion of dECM microparticles is an alternative, safe and effective technique for obtaining homogenous particle size distribution. In addition, this methodology efficiently preserves the molecular constituents of native tissue which can help support a constructive, site-appropriate, remodeling response when implanted in vivo [29]. 

### 3.3. Characterization of Fibrin and Fibrin–dECM Hydrogels

Human fibrin sealant or glue (TISSEEL^®^) was employed as a vehicle for the delivery of dECM microparticles for scarless wound healing of cornea. The fibrin derived hydrogels were prepared with minor modifications to the manufacturer’s protocol, as depicted in Figure 3A. The physical and mechanical properties of the prepared fibrin-based hydrogels were evaluated using several techniques in vitro. The crosslinking kinetics of the prepared fibrin derived hydrogels was evaluated using a parallel plate rheometer. Our results show that the storage modulus of fibrin glue plateaued at 28 min, whereas fibrin–EdECM hydrogels attained saturation within 17 min (Figure 3B). This observation suggests that the time required for complete crosslinking of fibrin–EdECM hydrogels is significantly shorter compared to fibrin hydrogels, which should in turn reduce the postoperative recovery time of a patient in a clinical setting.

We next evaluated the compressive modulus of fibrin hydrogels using a mechanical testing instrument. The results demonstrate that fibrin hydrogels displayed significantly higher compressive strength (102.97 ± 23.95 kPa) compared to fibrin–EdECM hydrogels (70.83 ± 9.17 kPa) (Figure 3C). Although studies have shown that incorporation of micro-/nanoparticles improve the mechanical properties of fabricated hydrogels [30,31], surprisingly, the inclusion of EdECM microparticles in fibrin glue did not improve the compressive modulus of fibrin–EdECM hydrogels. Nevertheless, as the compressive modulus values of both hydrogel groups fall within the range of 25–100 kPa, reported to be the Young’s modulus of human cornea tissue [32,33], the engineered fibrin–EdECM hydrogels are favorably poised to support cell adhesion and promote stromal tissue regeneration in vivo.

Application of fibrin glue has been shown to be effective for treatment of corneal perforations [34]. Hence, to evaluate the sealing ability of fibrin–EdECM hydrogels, burst pressure assessment was performed on perforated cadaveric human corneas ex vivo. Maximum burst pressure values for fibrin controls were observed at 21.7 ± 4.3 kPa, whereas fibrin–EdECM hydrogels sustained significantly higher burst pressures of 34.3 ± 3.7 kPa (Figure 3D). These data indicate that fibrin–dECM hydrogels possess sufficient adhesive strength to sustain pressures in the order of magnitude higher than the nominal intraocular pressure (IOP) of the human eye (~2 kPa) [35]. 

Furthermore, MTT assay on L929 cells incubated in fibrin and fibrin–dECM hydrogel extracts demonstrated negligible cytotoxicity with percentage cell viability ranging 82–97% compared to media control at all four tested concentrations (25%, 50%, 75%, and 100%) (Appendix A). These results show that the prepared fibrin-based hydrogels and their degradation products were non-cytotoxic to L929 cells in vitro. In addition, results from genotoxicity assay demonstrate that the mean number of revertant colonies at the concentration of polar and non-polar extracts of fibrin derived hydrogels was comparable to those of solvent controls, in both the presence and absence of metabolic activation. These data suggest that the fibrin derived hydrogel extracts were “non-mutagenic” to *S. typhimurium* and *E. coli* strains at the concentration of 100% extract solution as assessed via bacterial reverse mutation test (Appendix A).

### 3.4. In Vitro hCSC Culture Studies

To demonstrate the cytocompatibility of EdECM microparticles, hCSCs were encapsulated and cultured in fibrin–EdECM hydrogels for a period of five days in vitro. The results from the live–dead assay demonstrate that the EdECM microparticles displayed very good cytocompatibility, comparable to fibrin-only controls, with cell viability exceeding >95% by the end of Day 5 (Figure 3E). Next, to study the influence of biological-derived EdECM microparticles on the phenotype of hCSCs encapsulated in fibrin hydrogels, cells were stained for specific mesenchymal stem cell (MSC) markers, CD73 and CD90. Of note, CD73 and CD90 are common stem cell markers that are routinely employed for the identification of MSCs arising from various tissue types in the body [36]. The results from immunofluorescence studies show that hCSCs cultured in fibrin–EdECM hydrogels exhibited significantly higher expression of CD73 and lower of CD90 compared to fibrin controls at the end of the five-day culture period (Figure 3F,G and Appendix A). In addition, the fibrocytic marker α-SMA, a standard marker used for labeling activated fibrocytes/myofibroblasts that play a major role in tissue fibrosis [37], was significantly downregulated in cells cultured in the presence of EdECM microparticles compared to fibrin controls (Figure 3H and Appendix A) at all three time points. This data clearly indicate that EdECM microparticles inhibit the differentiation of hCSCs to a myofibroblast lineage which plays a major role in fibrotic ECM deposition that leads to corneal opacity [38]. Overall, these data demonstrate that dECM microparticles support and maintain hCSC phenotypic characteristics that are beneficial for tissue repair and regeneration in vivo.

Biodegradation studies on cell-laden fibrin-based hydrogels showed that both hydrogel groups lost only 20% of their mass by the end of Day 6 (Figure 3I). However, both groups significantly lost >80% mass by the end of Day 8. Moreover, no significant differences in percentage mass loss were observed between the two groups at any time point during the eight-day study period. These data correlate well with other studies that show that fibrin gels possess weak physical properties and hence are associated with a fast degradation rate [5,39].

### 3.5. Animal Studies

For in vivo studies, physically processed dECM microparticles were employed in the hydrogel preparation process as enzymatic digestion of physically milled dECM powder generated very low yield of EdECM microparticles. Skin sensitization studies with the polar and non-polar fibrin-based hydrogel extracts demonstrated that no treatment related skin reactions were observed at sites after intradermal injection. However, skin reactions such as erythema (of varying degree) and very slight edema (barely perceptible) were observed at all other injection sites in all animal groups. In topical induction and challenge phases, no treatment-related skin reactions such as erythema and edema were observed in any of the animals of all groups tested. Cumulatively, our results indicate that extracts of fibrin and fibrin–dECM hydrogels were found to be “non-sensitive” to the skin of Guinea pigs under the employed experimental conditions (Appendix A).

Ocular irritation studies demonstrated that no treatment related ocular lesions were observed for up to 72 h in initial and confirmatory test animals after single ocular instillation of polar hydrogel extracts (Appendix A). In addition, no treatment related clinical signs of toxicity, mortality, and gross pathological changes were observed in any of the animals in both initial and confirmatory test groups, post-instillation. Taken together, based on the observed results under the experimental conditions followed as per ISO 10993 guidelines, it can be concluded that the fibrin derived hydrogel extracts were found to be a “non-irritant” to the eyes of New Zealand white rabbits. 

We next assessed the use of fibrin-based hydrogels in a rabbit model of corneal stromal injury. Animals that received hydrogel formulations did not exhibit adverse reactions but displayed moderate ocular irritation and some mucus release for the initial 2–3 days post-surgery. However, there was no sign of edema, opacity or neo vascularization in the cornea and no significant differences in IOP were observed between the experimental and normal eyes of rabbits in all four groups. Clinical photographs demonstrated that a clear demarcation of the wound site was visible in all animals at the end of Day 7 (data not shown). Cobalt blue slit lamp photographs with fluorescein staining demonstrated that all four groups re-epithelialized by Day 14 post-surgery, implying successful migration of corneal epithelial cells over the fibrin-based hydrogels (Figure 4B). However, corneal healing and scar formation were also observed, but their magnitude varied across all four groups. For instance, slit lamp images of untreated corneas revealed severe scar formation that was visible until Day 14 but gradually decreased over the next few weeks. On the other hand, corneas treated with the various treatment formulations demonstrated a gradual decrease in corneal haze that was no longer visible after four weeks. 

ASOCT is an established platform for evaluating corneal thickness and has been used to identify and evaluate changes occurring along the corneal surface non-invasively. ASOCT imaging demonstrated that all animals exhibited nominal corneal thickness with no pathological conditions prior to surgery. After surgery, ASOCT images showed a reduced corneal thickness with an epithelial defect measuring 150–200 µm depth. Interestingly, one week post-surgery, corneas that received fibrin hydrogels with dECM microparticles retained the bioadhesive more prominently compared to fibrin-only controls, suggesting that dECM microparticles played an important role in stabilizing the hydrogels in vivo (data not shown). In addition, ASOCT imaging also indicated scar formation, visible as a hyper-reflective layer due to incident light scatter from the underlying stromal layer, with varying degrees along with signs of re-epithelialization in all four groups across all time points during the eight-week study period.

Corneal densitometry has been employed clinically to quantify corneal haze and to determine the extent of scarring in the cornea. Prior to surgery, animals demonstrated nominal densitometric values (~20%), which was suggestive of good optical clarity and absence of deformities in the cornea. However, post-surgery, densitometric scans indicated high light scattering, similar to ASOCT imaging, which is indicative of scar development in the injured eye of the rabbit. Corneal opacity was close to 70% across all groups during the first two weeks post-surgery, following which the values gradually decreased, suggesting wound stabilization and healing at the injured site (Figure 4C). By the end of eight weeks, densitometric values dropped to ~45%, suggesting gradual recovery of vision in the injured eye of the rabbits. It was observed that animals that received fibrin with dECM microparticles exhibited the lowest corneal opacity values at the eight-week time point, suggesting that the presence of dECM might have minimized corneal stromal scarring compared to the other groups. Corneas that received fibrin hydrogels with dECM microparticles and hCSCs displayed slightly higher densitometric values compared to other three groups.

Histopathological evaluation demonstrated that corneas from all four groups showed evidence of re-epithelialization and stromal reconstruction (Figure 5). More importantly, histological sections revealed strong adhesion of the fibrin-based hydrogels at the defect site after application. It was observed that corneas that received fibrin + dECM microparticles showed the presence of voids due to the absence of microparticles in the fibrin hydrogel. It is not clear whether the microparticles degraded overtime in vivo or were lost during histological sample processing and preparation. In contrast, corneas that received fibrin hydrogel with microparticles and hCSCs displayed a compacted epithelium at the injured site. However, epithelial hyperplasia due to the injury was visible across all four groups. PAS staining exhibited no sign of goblet cells, which suggests that there was no infiltration in cornea. Immunofluorescence imaging demonstrated a positive stain for the epithelial marker CK3. This result is noteworthy as cytokeratins, CK3 and CK12, are widely known to be expressed in differentiated human corneal epithelial cells [40].

Overall, our preclinical pilot study demonstrates that human cornea-derived ECM microparticles have an excellent safety profile and suggests that they can be potentially used for the treatment of surface epithelial defects and anterior stromal injuries in a minimally invasive manner. As the animal study performed was based on a pilot study design with very few animals, the efficacy of the various treatment formulations via standard statistical methods of analysis could not be established. In our previous study using Bovine DCM hydrogel, we showed its advantage over existing materials including easy availability, affordability, and simple formulation procedures, which make it a promising biomaterial to prevent scar development in an injured cornea [41]. Moreover, based on the data in hand, we are in the process of refining the stromal wound model as our results indicate that epithelial/stromal defects in rabbits heal rapidly as untreated corneas exhibited accelerated wound healing and speedy recovery without any therapeutic intervention. In view of this, studies have been focused on developing delayed wound healing models that mimic persistent corneal epithelial defects observed in several chronic ocular pathologies [21,42,43]. To overcome the above limitation and for plausible assessment of treatment outcomes, we are in the process of establishing and validating an alkali-induced stromal wound model for subsequent preclinical investigations. 

## 4. Conclusions

In this study, we report the use of dECM microparticles derived via physical and enzymatic processing of cadaveric human corneas as an accessible therapeutic option for the reconstruction of corneal surface post-injury. dECM microparticles averaged <10 µm in size and were readily dispersible in the precursor solution of thrombin. Hydrogels derived from fibrin–EdECM formulations were moderately transparent and exhibited physical and mechanical properties that matched the microenvironment of native cornea tissue. More specifically, fibrin–EdECM hydrogels exhibited good compressive modulus and sustained pressures 17-fold higher than the nominal IOP of the human eye. Furthermore, fibrin–dECM hydrogels demonstrated excellent biocompatibility, non-mutagenicity, and, in addition, was found to be non-sensitive and non-irritable to the ocular surface as assessed via in vivo studies following ISO 10993 guidelines. Moreover, the prepared hydrogels supported the growth and maintained the phenotype of encapsulated hCSCs in vitro. Most importantly, fibrin–dECM hydrogels demonstrated safety and promoted corneal re-epithelialization and stromal regeneration in a rabbit model of anterior lamellar keratectomy in vivo. However, the limitation of our study is the small animal number in each group, and this needs to be further explored in a large preclinical study using both mice and rabbit scar models. Viewed comprehensively, our results indicate that dECM microparticles hold great promise for inducing constructive corneal remodeling for reparation of epithelial surface defects and anterior stromal wounds. Evidently, this methodology should reduce the dependence on donor corneas for full corneal transplant (keratoplasty) and circumvent issues associated with current corneal transplantation procedures, making it a viable, minimally invasive approach that will lead to faster recovery, improved visual acuity, minimized photosensitivity to light, reduced pain, and better the quality of life of patients.

## Figures and Tables

**Figure 1 biomolecules-11-00532-f001:**
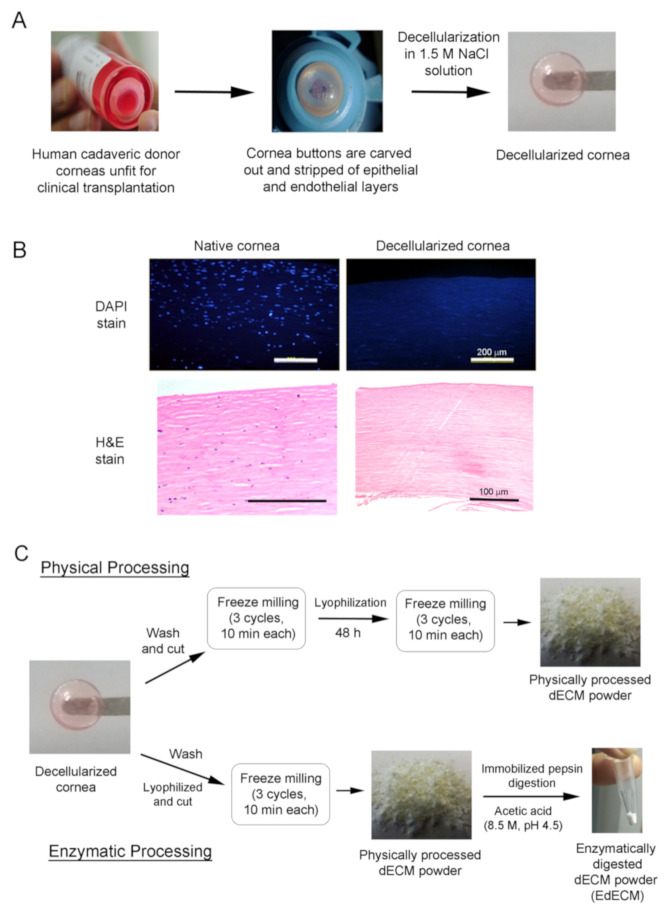
Schematic representation depicting the preparation of physically milled and enzymatically digested human cadaveric cornea derived decellularized ECM microparticles. (**A**) Decellularization process of cadaveric human donor corneas unfit for clinical transplantation; (**B**) nuclei (DAPI) and H&E stain demonstrating the absence of genetic material; and (**C**) preparation of physically processed and pepsin enzyme digested dECM microparticles from decellularized human corneas.

**Figure 2 biomolecules-11-00532-f002:**
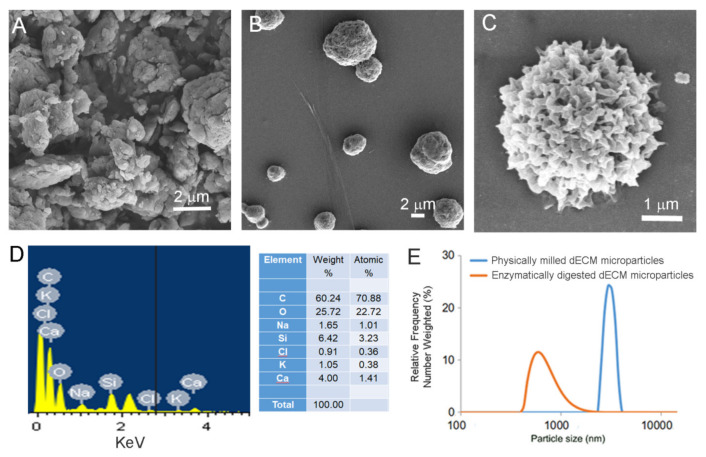
Physical characterization of dECM microparticles: (**A**) representative SEM image of physically processed dECM microparticles; (**B**) SEM image of enzymatically digested dECM microparticles; (**C**) SEM images of a EdECM microparticle showing petaloid-like architecture; (**D**) elemental analysis of EdECM microparticles demonstrating that its primarily composed of carbon and oxygen moieties confirming its ECM origin; and (**E**) hydrodynamic sizes of dECM microparticles via DLS measurements.

**Figure 3 biomolecules-11-00532-f003:**
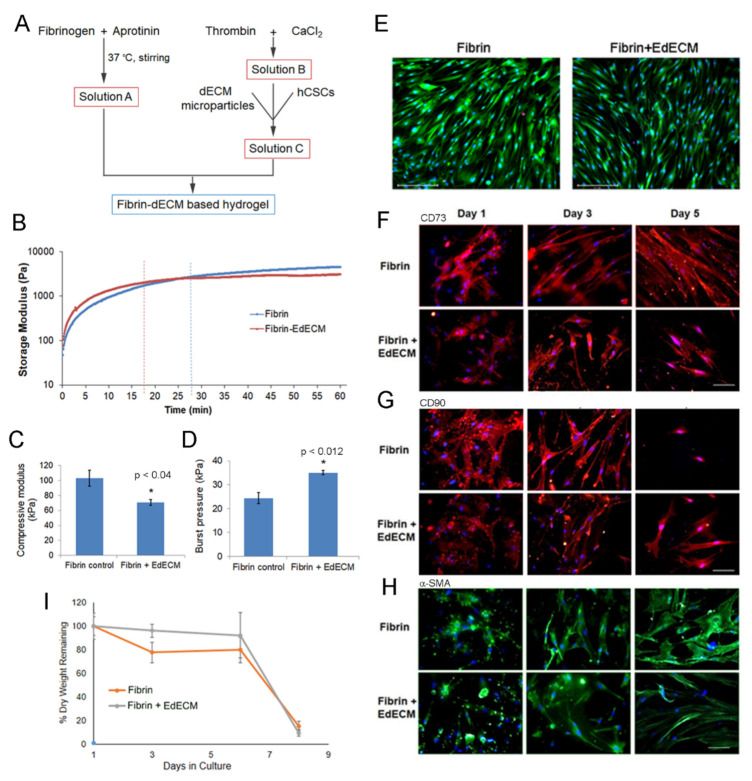
Characterization of fibrin–EdECM based hydrogels: (**A**) schematic for the preparation of fibrin–dECM based hydrogels; (**B**) crosslinking kinetics of fibrin and fibrin–EdECM hydrogels; (**C**) compressive modulus of fibrin and fibrin–EdECM hydrogels; (**D**) maximum burst pressure of fibrin and fibrin–EdECM hydrogels; (**E**) cell viability of hCSCs encapsulated in fibrin and fibrin–EdECM hydrogel on Day 5 via live/dead stain, where reen denotes live cells and red denotes dead cells and nuclei are labeled blue; (**F**,**G**) CD73 and CD90 biomarker expression of hCSCs encapsulated in fibrin and fibrin–EdECM hydrogels; (**H**) expression of α-SMA in hCSCs encapsulated in fibrin and fibrin–EdECM hydrogels; and (**I**) biodegradation of hCSC encapsulated fibrin and fibrin–EdECM hydrogels in vitro. Data are represented as mean ± SE with n≥3 samples/group. * *p* ≤ 0.05 denotes significant differences observed between fibrin and fibrin–EdECM hydrogels.

**Figure 4 biomolecules-11-00532-f004:**
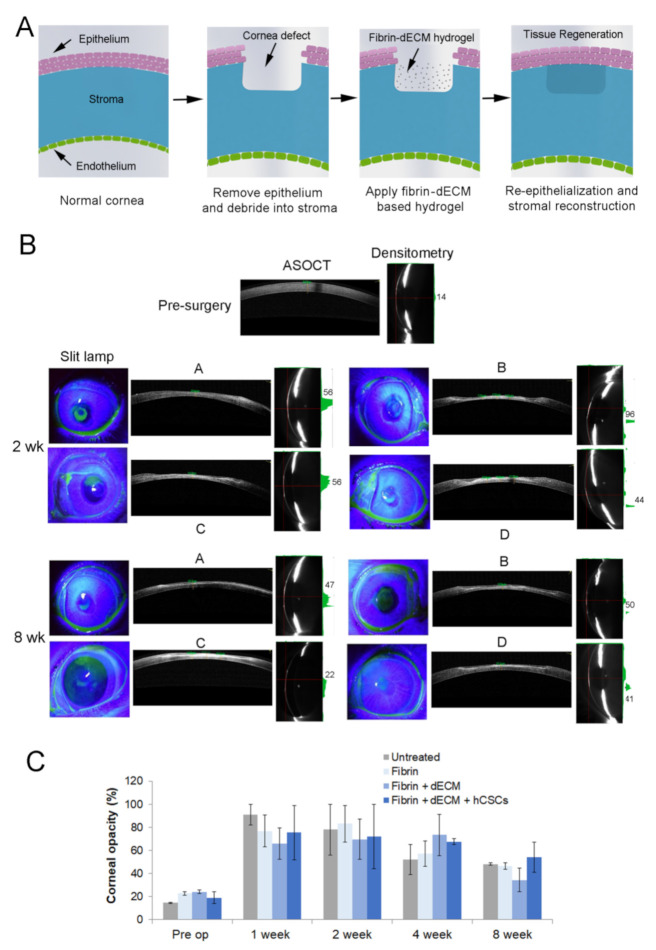
Rabbit cornea imaging in vivo. (**A**) Schematic depicting the creation of corneal defect and application of fibrin–dECM based hydrogels for ocular tissue reconstruction. (**B**) Representative images of rabbit corneas obtained using slit lamp with fluorescein staining, ASOCT and densitometry two and eight weeks post-application of fibrin based hydrogels. Corneas are represented as: (A) untreated; (B) fibrin controls; (C) fibrin + dECM microparticles; and (D) fibrin + dECM + hCSCs. Prior to surgery, corneas of all animals exhibited nominal corneal thickness with good optical clarity. (**C**) Bar graph depicts the decrease in corneal haze, an indicator of corneal wound healing, determined via densitometric evaluations during the eight-week time period.

**Figure 5 biomolecules-11-00532-f005:**
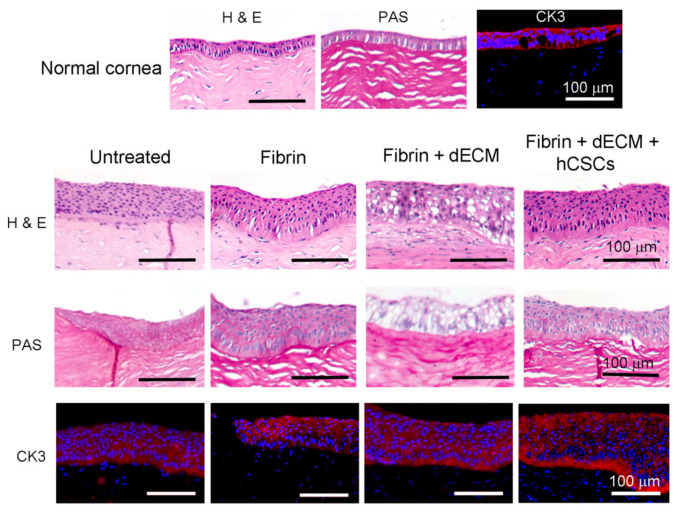
Histopathology and immunofluorescence imaging of paraffin-embedded rabbit cornea sections. H&E imaging revealed strong adhesion of the fibrin-based adhesives to the cornea and demonstrated evidence of re-epithelialization and stromal reconstruction with signs of epithelial hyperplasia. PAS staining denoted the absence of goblet cell infiltration in the cornea. Immunofluorescence staining demonstrated a positive stain for cytokeratin 3, observed across all groups, confirming corneal re-epithelialization by the end of eight weeks.

**Table 1 biomolecules-11-00532-t001:** List of formulations used for the various characterization studies.

dECM Microparticle Characterization	
**SEM**	**Physically processed dECM and EdECM**
Thermogravimetric analysis (TGA)	Physically processed dECM
Dynamic Light Scattering (DLS)	Physically processed dECM and EdECM
Fourier Transform Infrared (FTIR)	Physically processed dECM and EdECM
Sandwich ELISA assay	EdECM
Mass Spectrometry	EdECM
**Hydrogel Characterization**	
Compressive modulus	EdECM
Crosslinking kinetics	EdECM
Ex-vivo burst pressure	EdECM
**In vitro Cell Culture Studies with hCSCS**	
MTT assay on hydrogel extracts	Physically processed dECM
Bacterial reverse mutation test on hydrogel extracts	Physically processed dECM
Cell encapsulation in hydrogels	EdECM
Live/Dead assay	EdECM
Biomarker expression	EdECM
Biodegradation Study in vitro	EdECM
**In Vivo Studies**	
Skin sensitization test in Guinea pigs	Physically processed dECM
Acute ocular irritation test in rabbits	Physically processed dECM
Treatment of corneal stromal injury in rabbit model	Physically processed dECM

## Data Availability

The data presented in this study are available on request from the corresponding author. The data are not publicly available due to intellectual property restrictions and institutional policies.

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
