# Peer review of "Human Cadaveric Donor Cornea Derived Extra Cellular Matrix Microparticles for Minimally Invasive Healing/Regeneration of Corneal Wounds"

_biomolecules, 2021, doi:10.3390/biom11040532_

Round 1
Reviewer 1 Report
In the submitted paper by Chandru et al, the authors developed a human cornea derived from decellularized extracellular matrix (dECM) microparticles mixed with fibrin hydrogels as an alternate application for corneal wound healing. The study showed successful decellularization of the donor cornea. The authors also compared and characterized the mechanically processed dECM and pepsin enzyme digested ECM (EdECM). The results indicated that enzyme digested ECM retained all the characteristics of mechanically processed dECM, and further insignificant amount of pepsin was observed in the end product of EdECM. The authors then compared the wound healing properties in vivo between the existing fibrin hydrogels alone, fibrin gels plus dECM, and fibrin gels plus dECM mixed with human corneal stem cells (hCSCs). These comparisons showed that hydrogels derived from fibrin-dECM demonstrated excellent biocompatibility, non-mutagenicity, and was also found not to induce sensitivity or irritation on the ocular surface. Further, fibrin-dECM hydrogels promoted corneal re-epithelialization and stromal regeneration in a rabbit model of anterior lamellar keratectomy.
This is a very nice study performed comprehensively, and is a promising and minimally invasive future procedure to treat epithelial and stromal wounds.
Minor comments:
Please perform a spelling check. In the abstract, decellularized is spelled incorrectly.
Introduce the definition for the acronym (dECM) in the abstract.
Change Suppl. Figure 1A, Suppl. Figure 1B, Suppl. Figure 1C to Figure S1, Figure S2 and Figure S3 respectively.
There was no mentioning of Figure 3C in the text.
Major Comments:
Figure 3: F and G – It would be wise to label these immunofluorescence images with CD73 and CD90, respectively.
As the authors claim that CD73 and CD90 staining of fibrin+EdECM was more intense than for fibrin alone, but the figure shows the contrary. The top panel of fibrin has stronger staining for CD73 than the fibrin+EdECM, lower panel. The CD90 staining however appears less for fibrin alone only at day 5 compared to fibrin+EdECM. This experiment would be better represented if a fluorescence quantification was performed.
A major concern is that while the biochemical and cell culture characterization was done using EdECM, the in vivo experiments were conducted using physically processed dECM. Understandably, the authors mention that there is low yield of EdECM. However, at minimum EdECM should be shown in the ocular irritation studies to be safe?
Author Response
We thank you for giving us chance to review our manuscript. We have addressed all the points raised by the reviewers. The respective modifications/corrections are quoted in the attached manuscript file (word document, provided with line numbers in track mode). Please find the response for your comments down below.
Hope the paper may be accepted in revised format.
Pointwise Response to Comments
MINOR COMMENTS
Point 1: Please perform a spelling check. In the abstract, decellularized is spelled incorrectly.

The typo has been corrected in line22, page 1.
Point 2: Introduce the definition for the acronym (dECM) in the abstract.
The acronym for “dECM” has been mentioned in line 22, Page 1.
Point 3: Change Suppl. Figure 1A, Suppl. Figure 1B, and Suppl. Figure 1C to Figure S1, Figure S2 and Figure S3 respectively.
The changes have been included in the manuscript text as well as the supplementary document.
Point 4: There was no mentioning of Figure 3C in the text.
“Figure 3C” has now been introduced in the text at line 553, page 14.
MAJOR COMMENTS
Point 5: Figure 3: F and G – It would be wise to label these immunofluorescence images with CD73 and CD90, respectively.
The IF images have been labelled as CD73 and CD90 to differentiate the two images that are labelled with the same secondary antibody.
Point 6: As the authors claim that CD73 and CD90 staining of fibrin+EdECM was more intense than for fibrin alone, but the figure shows the contrary. The top panel of fibrin has stronger staining for CD73 than the fibrin+EdECM, lower panel. The CD90 staining however appears less for fibrin alone only at day 5 compared to fibrin+EdECM. This experiment would be better represented if a fluorescence quantification was performed.
Quantification of fluorescence signals was performed for the three biomarkers with at least 15 ROIs per group from 3 different images and the data has been added to the supplementary information as Figure S5. The results have also been modified in section 3.4 to reflect the same (lines 585-592, Page 16).
Point 7: A major concern is that while the biochemical and cell culture characterization was done using EdECM, the in vivo experiments were conducted using physically processed dECM. Understandably, the authors mention that there is low yield of EdECM. However, at minimum EdECM should be shown in the ocular irritation studies to be safe?
We agree that at minimum ocular irritation studies should have been performed with EdECM microparticles. However, as EdECM preparation gave us very low yields and coupled to the fact that ocular irritation studies are in vivo studies performed in rabbits by a certified Contract research organization (CRO), we did not pursue with these experiments.
Reviewer 2 Report
Authors presented in this manuscript an extensive description of the production of cornea derived extracellular matrix microparticles of human origin. The problem of corneal donors unavailability is determinant for clinical practice and global ocular health. The strategy presented here is original and potentially useful if data supported the hypothesis.
Introduction is well written and provides a clear view of the problem and proposed solution.
Authors have used a large variety of techniques to evaluate their products. Although Methods section is well organized, many issues must be amended.
Along the manuscript it is not clear how many different formulations were obtained and used in each case. It would be helpful to include a table or a brief list of formulations in methods section as some of the formulatios were used for one experiment and some to others.
Authors must detail the model and brand of each equipment they have used, including freeze-miller, X-ray spectroscopy, columns or microplate reader. Also they must provide information about reactives (brands).
In section 2.1 it would be useful to know how many corneas were used in this work and how many were destined to each fabrication procedure. Also it should be of help if authors describe if they used the complete cornea or removed epithelium and/or endothelium prior to decellularization or if they needed a pool of cornea buttons to obtain the lyophilized product. A measure of initial and final weight may help. Authors must detail the immunostaining procedures carried out to make sure no cellular debris was present. DAPI staining (figure 1.B) is not an immunolabeling technique as it is not based on antibody to antigen recognition. It is a fluorescent histochemical reaction. This technique does not give any information about cellular debris as it is restricted to genetic material. Please correct the legend at figure 1.
Additional means to ensure elimination of cellular debris must be presented in order to validate total decellularization.
In section 2.2, there is no data supporting the idea of water acting as grinding agent during freeze-milling. It is just a supposition and must be removed.
It must be included in Methods section the number of samples used in each biophysical tests, MTT assay or in vitro experiments. How many corneal rims did you use to obtain hCSCs? Which is the cellular yield from each rim?
In section 2.6.4, authors must explain briefly the procedure of immunofluorescence staining, i.e. duration of primary and secondary antibodies incubations, blocking agents (if used), antigen retrieval (if needed), etc.
At the beginning of section 2.7 I would include the ethic statements and guidelines. Also, a detailed explanation of sample size (number of animals in each group) could help to better understand the results. For example, it is suggested to include the number of rabbits used in acute irritation test (n=3?) even when it is regulated by the ISO rule.
It is required an explanation of the distribution of each of the 9 rabbits at the in vivo injury model into the four groups of treatment (n=2?). Authors must detail if control group is injured or uninjured and provide evidences of the effect of each treatment in uninjured eyes.
Authors must detail the duration and dose of treatments. Did you treat only once after injury? Or were it repeated applications? It must be discussed in depth how the differences on wound dimensions could affect to the progression of regeneration.
In general, results are incomplete and many experiments were made only with one set of formulations (fibrin and fibrin-dECM in MTT assay versus fibrin-EdECM hydrogels for burst pressure, for example). It is required a clear explanation of the criteria to use one or another formulation apparently randomly.
In section 3.3 authors affirm that fibrin-EdECM hydrogels sustained significantly higher burst pressures but they don’t provide the statistical p value.
In section 3.4, authors affirm that their data clearly indicates that EdECM inhibit the differentiation of hCSCs to myofibroblasts even when they didn’t make any quantitative analysis of CD73 and CD90 expression or αSMA downregulation. Images from figure 3H seem to have similar intensity and the distribution of the staining is not representative. Authors must provide quantitative or semiquantitative data to make that affirmation. It is required at least the number of samples observed of each group.
As I can understand, one of the advantages of dECM microparticles is that decellulariztion makes useful to use allogenic cadaveric corneas to treat corneal injuries or to improve surgical procedures. However, this advantage is diluted when combined with an allogenic source of hCSCs. I would prepare two different works with these formulations. One work regarding dECM use as corneal glue additive to highlight the goodness of biocompatibility and no host tissue reaction. Another, including the use of hCSCs as a therapy to treat corneal injuries, in combination with dECM as a bulk matrix to improve regeneration. In other case, a new redaction of the manuscript may be done in order to focus the goal in a combined treatment to promote corneal tissue regeneration.
What is the benefit of dECM microparticles respecting fibrin hydrogel? It seems not to be statistical differences between treatments.
It is recommended to obtain some better immunofluorescence and histology images and to complete the study with immunostaining of αSMA myofibroblasts on the stroma on equivalent sections. This will support results from ASOCT.
In my opinion this is a nice idea and a pretty good strategy to overcome the lack of transplant tissue and hard work have been done but the inconsistence of some of the results and the lack of clear statistical differences between treatments (and between treatments and untreated groups) must be corrected with additional results.
Author Response
We thank you for giving us chance to revise our manuscript. We have addressed all the points raised by you. The respective modifications/corrections are quoted in the attached manuscript file (word document, provided with line numbers in track mode). We have enclosed a single file that includes the amended manuscript, supplementary data.
Pointwise Response to all Comments
Point 1: Along the manuscript it is not clear how many different formulations were obtained and used in each case. It would be helpful to include a table or a brief list of formulations in methods section as some of the formulations were used for one experiment and some to others.
To address the above concern, Table 1 titled “List of formulations used for the various characterization studies” has been inserted on page 11 after section 2.10.
Point 2: Authors must detail the model and brand of each equipment they have used, including freeze-miller, X-ray spectroscopy, columns or microplate reader. Also they must provide information about reactives (brands).
Freeze miller - Spex SamplePrep, USA; 6775-230; line 137, Page 5
Freeze dryer – Labcogene, Scanvac CoolSafe Pro #110-4, line 140, page 5
X-ray spectroscopy - Zeiss Ultra-55, Germany; line 162, page 5
Microplate reader - Perkin Elmer, EnSpire®, USA; line 247, page 7
MTT – HiMedia, India; line 244, page 7
Formalin solution – Sigma-Aldrich, India; line 307, page 8
Salmonella typhimurium strains TA98, TA100, TA1535, TA1537 and Escherichia coli WP2 uvrA (pKM101) - Molecular Toxicology Inc., USA; line 264, page 7
Point 3: In section 2.1 it would be useful to know how many corneas were used in this work and how many were destined to each fabrication procedure. Also it should be of help if authors describe if they used the complete cornea or removed epithelium and/or endothelium prior to decellularization or if they needed a pool of cornea buttons to obtain the lyophilized product. A measure of initial and final weight may help. Authors must detail the immunostaining procedures carried out to make sure no cellular debris was present. DAPI staining (figure 1.B) is not an immunolabeling technique as it is not based on antibody to antigen recognition. It is a fluorescent histochemical reaction. This technique does not give any information about cellular debris as it is restricted to genetic material. Please correct the legend at figure 1.
Corneas stripped of the epithelium and endothelium were lyophilized and freeze milled to yield dECM powder. This change is reflected in line 112, page 3.
The phrase “and cell remnants” is removed from line 127, page 4.
180 corneas were used for this work during the course of the study towards decellularization and generating dECM powder from them.
The weight of corneas before lyophilization was found to be ~110 mg and the weight of dECM obtained from each cornea was 7-8mg.
Point 4: Additional means to ensure elimination of cellular debris must be presented in order to validate total Decellularization.
We too agree with the reviewer that we need to do other assays too. But we did not perform it in current studies. However in our previous study which was recently published (ACS Appl. Bio Mater. 2021, 4, 1, 533–544), we did biochemical analysis of the decellularized matrix and compared to that of the native tissue to determine the extent to which ECM components are retained, such as collagen and sGAGs. Our previous biochemical results from stabilised decellularisation method revealed that around 80.9 ± 8.44 and 33.4 ± 4.7% of sGAGs and total collagen were retained, respectively, in the decellularized tissue compared to that of the native corneal tissue.
Point 5: In section 2.2, there is no data supporting the idea of water acting as grinding agent during freeze-milling. It is just a supposition and must be removed.
We have in-house data to show that average particle size of freeze milled lyophilized corneas demonstrated higher particle size compared to particles obtained from direct milling of wet corneas. This data has been included in the supplementary section as Figure S1. Correspondingly, lines 132-135, pages 4-5, have been modified to reflect this result.
Point 6: It must be included in Methods section the number of samples used in each biophysical tests, MTT assay or in vitro experiments. How many corneal rims did you use to obtain hCSCs? Which is the cellular yield from each rim?
The no. of replicates used for each biophysical test is mentioned in the corresponding figure legends. A minimum of three replicate samples were used for each biophysical test. For MTT assay 6 replicate samples were used for the study (line 232, page 7). Furthermore, on an average one cornea rim provided ~8-10 million hCSCs at passage 3 and around 3-5 corneas were used for in vitro cell culture studies (lines 275-276, page 7).
Point 7: In section 2.6.4, authors must explain briefly the procedure of immunofluorescence staining, i.e. duration of primary and secondary antibodies incubations, blocking agents (if used), antigen retrieval (if needed), etc.
Page 8, lines 312-318; the detailed immunofluorescence staining protocol has been added to section 2.6.4.
Point 8: At the beginning of section 2.7 I would include the ethic statements and guidelines. Also, a detailed explanation of sample size (number of animals in each group) could help to better understand the results. For example, it is suggested to include the number of rabbits used in acute irritation test (n=3?) even when it is regulated by the ISO rule.
As suggested, the ethic statements and guidelines related to animal studies are listed directly in section 2.7. Lines 331-344, pages 9, “Skin sensitization test in Guinea pigs and acute ocular irritation test in rabbits were per-formed in an AAALAC accredited facility in accordance with the recommendation of the Committee for the Purpose of Control and Supervision of Experiments on Animals (CPCSEA) guidelines for laboratory animal facility published in the Gazette of India, 2018, in accordance with the protocol approved by Institutional Animal Ethics Committee (IAEC) (Protocol Nos. BIO-IAEC- 3579 & BIO-IAEC-3385) and in accordance with the International Standard ISO 10993 Second Edition: 2006-07-15, “Biological Evaluation of Medical Devices - Part 2: Animal Welfare Requirements”. Reference Number: ISO 10993-2:2006(E).
The corneal stromal injury in rabbit model was conducted after IRB approval from the ethics committee at L V Prasad Eye Institute and animal Ethics committee of Vimta Labs Limited in accordance with the tenets of declaration from Helsinki (Ethics Ref: LEC-12-19-371).”
The no. of animals for each of the test study performed as per ISO guidelines has been explicitly mentioned under the relevant sections. The results have also been included in the supplementary section of the manuscript.
Point 9: It is required an explanation of the distribution of each of the 9 rabbits at the in vivo injury model into the four groups of treatment (n=2?). Authors must detail if control group is injured or uninjured and provide evidences of the effect of each treatment in uninjured eyes.
Lines 386-388, Page 10; “Animals were divided in to the following four groups. These were untreated control (2 rabbits), fibrin (2 rabbits), fibrin + EdECM (30 mg/mL) (3 rabbits), and fibrin + EdECM + hCSCs (5x106 cells/mL) (2 rabbits).”
Lines 391-393, page 10; “Animals that underwent surgery but received no treatment were used as untreated controls and the healthy contralateral eyes of all rabbits were used as normal experimental controls.”
Point 10: Authors must detail the duration and dose of treatments. Did you treat only once after injury? Or were it repeated applications? It must be discussed in depth how the differences on wound dimensions could affect to the progression of regeneration.
Only one dose of the treatment formulation was given to rabbits. This detail has been clarified in line 388, page 10.
The wound dimensions used in the study were 3 mm dia and 150-200 micron depth. Given these dimensions, the volume of the wound, assuming it’s close to a cylinder, is 3.14*r2* depth, which is equal to 1 -1.4 mm3.
Therefore, technically, a volume of approx. 2 µL is required on an average for each rabbit. However, excess volume of the treatment formulation, ~4-8 µL, was dispensed according to the clinician’s discretion to account for the transfer loss in the pipette and to completely cover the wound of the rabbit. We do not anticipate significant changes in the progression of corneal regeneration as the wound dimensions fall under “superficial – anterior stromal injury”.
As per ethical approvals we were not able to perform many formulations and dose/ time kinetics as part of this projects. But in future studies we have plan to use our dECM in single and multiple doses too to understand better efficacy and safety too.
Point 11: In general, results are incomplete and many experiments were made only with one set of formulations (fibrin and fibrin-dECM in MTT assay versus fibrin-EdECM hydrogels for burst pressure, for example). It is required a clear explanation of the criteria to use one or another formulation apparently randomly.
Most in-house experiments detailed in Table 1 were performed with EdECM microparticles. However, for other in vitro and in vivo studies, including MTT assay, bacterial reverse mutation test, skin sensitization test, acute ocular irritation test and treatment of corneal stromal injury in rabbit model, physically processed dECM powder was used as it was performed following ISO standards through a government approved contract Research organisation (CRO). Hence, large quantities of dECM microparticles were needed for each in vivo study which was only achievable through physical processing of the decellularized corneas.
As per ethical approvals we were not able to perform many formulations and dose/ time kinetics as part of this projects. But in future studies we have plan to use our dECM in single and multiple doses too to understand better efficacy and safety too both in vivo and in vitro too.
Point 12: In section 3.3 authors affirm that fibrin-EdECM hydrogels sustained significantly higher burst pressures but they don’t provide the statistical p value.
Thanks for pointing it. The p values for compressive modulus (p ≤ 0.04) and burst pressure (p ≤ 0.012) have been added in Figure 3C and 3D, respectively.
Point 13: In section 3.4, authors affirm that their data clearly indicates that EdECM inhibit the differentiation of hCSCs to myofibroblasts even when they didn’t make any quantitative analysis of CD73 and CD90 expression or αSMA downregulation. Images from figure 3H seem to have similar intensity and the distribution of the staining is not representative. Authors must provide quantitative or semiquantitative data to make that affirmation. It is required at least the number of samples observed of each group.
Quantification of fluorescence signals was performed for the three biomarkers with at least 15 ROIs per group from 3 different images and the data has been added to the supplementary information as Figure S6. The results have also been modified in section 3.4 to reflect the same (lines 585-592, Page 16). Hope this additional data will give better understanding.
Point 14: As I can understand, one of the advantages of dECM microparticles is that decellulariztion makes useful to use allogenic cadaveric corneas to treat corneal injuries or to improve surgical procedures. However, this advantage is diluted when combined with an allogenic source of hCSCs. I would prepare two different works with these formulations. One work regarding dECM use as corneal glue additive to highlight the goodness of biocompatibility and no host tissue reaction. Another, including the use of hCSCs as a therapy to treat corneal injuries, in combination with dECM as a bulk matrix to improve regeneration. In other case, a new redaction of the manuscript may be done in order to focus the goal in a combined treatment to promote corneal tissue regeneration.
We completely agree with the above rational. However, as the in vivo work in this study was looked upon as a pilot, we restricted our work to only 2-3 animals. Based on the learnings and outcome of this work, we are planning to go ahead with a power study with the four animal groups to demonstrate efficacy of the various treatment formulations with respect to corneal regeneration via standard statistical methods of analysis in future.
Point 15: What is the benefit of dECM microparticles respecting fibrin hydrogel? It seems not to be statistical differences between treatments.
Although we did not see statistical differences between the various treatment options as our study was inherently designed as a pre-clinical pilot with very few animals, recent works on the use of decellularized ECM for promotion of corneal regeneration have shown significant therapeutic efficacy. For instance, Yin et al., has shown that the incorporation of cornea dECM microparticles in fibrin glue significantly improved cornea repair and reduced haze and scarring of the treated corneas.1 Similarly, compressed collagen intermixed with cornea-derived dECM has shown good mechanical properties and biochemical niches comparable to native corneal stroma and demonstrated significantly higher expression of keratocyte-specific genes that are conducive for corneal regeneration.2
Refs:
- H. Yin, Q. Lu, X. Wang, S. Majumdar, A.S. Jun, W.J. Stark, M.P. Grant, J.H. Elisseeff, Tissue-derived microparticles reduce in-flammation and fibrosis in cornea wounds, Acta biomaterialia, 85 (2019) 192-202.
- Hong H, Kim H, Han SJ, Jang J, Kim HK, Cho DW, Kim DS. Compressed collagen intermixed with cornea-derived decellularized extracellular matrix providing mechanical and biochemical niches for corneal stroma analogue. Mater Sci Eng C Mater Biol Appl. 2019 Oct;103:109837.
Point 16: It is recommended to obtain some better immunofluorescence and histology images and to complete the study with immunostaining of αSMA myofibroblasts on the stroma on equivalent sections. This will support results from ASOCT.
We too agree and we tried it. But as there is no specific antibody against rabbit tissue, we did not get desired results. We may try this with other antibodies in future. It’s now beyond the scope of this project.
Point 17: In my opinion this is a nice idea and a pretty good strategy to overcome the lack of transplant tissue and hard work have been done but the inconsistence of some of the results and the lack of clear statistical differences between treatments (and between treatments and untreated groups) must be corrected with additional results.
Based on the learnings and outcome of this work, we are currently going ahead with a power study with the four animal groups to demonstrate efficacy of the various treatment formulations with respect to corneal regeneration via standard statistical methods of analysis.
Round 2
Reviewer 2 Report
Thank you for your replies to my inquiries. Authors have fulfilled successfully any controversial issue about their study. The corrections have improved significantly the reading and interpretation of the results and the work has clearly improved in quality.
The aim of this research line is important for the clinical practice in ocular surface surgery and I encourage authors to continue studying this promising approach.